# Diagnosis of Infectious Laryngotracheitis Outbreaks on Layer Hen and Broiler Breeder Farms in Vojvodina, Serbia

**DOI:** 10.3390/ani12243551

**Published:** 2022-12-15

**Authors:** Marko Pajić, Slobodan Knežević, Biljana Djurdjević, Vladimir Polaček, Dalibor Todorović, Tamaš Petrović, Sava Lazić

**Affiliations:** Scientific Veterinary Institute of “Novi Sad”, 21000 Novi Sad, Serbia

**Keywords:** laying hens, breeder flocks, infectious laryngotracheitis virus, respiratory symptoms

## Abstract

**Simple Summary:**

Infectious laryngotracheitis is a very important respiratory disease of poultry. It causes inflammation of the upper respiratory tract and mucous membranes of the eye. A large number of birds in the flock can be infected and mortality is variable. It is a viral disease, transmitted horizontally in a flock and causes great economic losses. In this study, we presented data on the occurrence of the disease after 20 years in unvaccinated flocks in the Vojvodina region and proposed a strategy for the prevention and control of this disease. Infected flocks had a severe clinical symptoms with a decrease in egg production and increased mortality. Clinical symptoms lasted from two to four weeks. We prevented the spread of the virus in one flock applying emergency vaccination with live vaccine. It can be concluded that the disease can easily spread between unvaccinated flocks. The most important thing is to improve biosecurity measures on the farm and implement vaccination in endemic areas.

**Abstract:**

Infectious laryngotracheitis (ILT) is a respiratory disease of poultry characterized by high morbidity and variable mortality. ILT is caused by *Gallid alpha herpesvirus-1* (GaHV-1), which is transmitted horizontally and most susceptible are chickens older than 4 weeks. After almost two decades since last appearance of this disease in Vojvodina, an outbreak occurred from April 2020 to August 2021 on five laying hen farms and one broiler breeder flock farm. Clinical signs were mild to severe respiratory symptoms, unilateral or bilateral head swelling, serous nasal discharge, conjunctivitis and increased tearing. There was a decrease in feed consumption (2.1–40.0%) and egg production (2.7–42.0%), weight loss and mortality increased (0.8–31.5%). Pathomorphological changes were localized in the upper respiratory tract. Total of 200 carcasses were examined; 40 pooled samples were analyzed by PCR, and 40 by bacteriological analysis. ILT virus was confirmed in tracheal tissue samples. Infected flocks were not vaccinated against this disease. Five flocks had coinfection with *Avibacterium paragallinarum*. Three-to-four weeks after the first reported case in the flock, clinical symptoms had ceased. Future control and prevention strategies will involve the procurement of flocks vaccinated by recombinant vaccine or the registration of live attenuated vaccines and their use during the rearing period.

## 1. Introduction

Infectious laryngotracheitis (ILT) is a highly contagious respiratory disease of domestic poultry caused by the herpes virus (*Gallid alphaherpesvirus* 1, fam. *Alphaherpesvirinae*) [1,2]. It is a DNA virus that infects the upper respiratory tract and conjunctiva, and the tracheal mucosa is affected the most [3,4]. It causes inflammation of the upper respiratory tract, heavy mucous discharge, cough and breathing difficulties, swelling of the infraorbital sinuses, decreased egg production and weight loss [5,6]. The disease is characterized by high morbidity (90–100%) and variable mortality (5–70%), most commonly from 10 to 20% [7]. Severe forms of the disease are characterized by severe dyspnea, expectoration of bloody mucus and high mortality due to suffocation [3,8]. The ILT virus can establish lifelong infections by establishing latency in the trigeminal ganglia. Stress factors such as the peak of egg production or transport may reactivate virus replication and excretion [6].

The disease is widespread and causes serious economic losses in poultry production [9,10]. Chickens of all ages can be infected, but those older than 4 weeks are most susceptible to infection [4]. The risk of disease in endemic areas is high. The virus is transmitted horizontally [10], while transovarian and vertical transmission through eggs have not been proven [8]. The sources of the ILT virus are infected chickens, latently infected chickens, dust, bedding, insects, drinking water and equipment. The ILT virus is able to persist in biofilms that can form in watering systems and thus be a source of infection for susceptible birds. Biofilm represents the accumulation of cells of microorganisms in the matrix of extracellular polymeric substances. In this way, microorganisms are resistant to some of the disinfectants [5]. It has been proven that insects of the *Alphitobius* genus on poultry farms can contain live viruses at least 42 days after the disease outbreak on the farm. Airflow between farms is one of the described indirect ways of spreading the ILT virus [4,5]. An epidemiologically important feature of this disease is that birds that have been vaccinated with live attenuated vaccines or have contracted the disease carry the virus in a latent form in the trigeminal ganglia. The virus can be reactivated in the stress phase (high stocking density, beginning of egg laying). The reactivated virus can cause the disease by being transmitted from bird to bird in susceptible populations, increasing its virulence [11,12]. Therefore, it is recommended that ILT live attenuated vaccines be used only in endemic areas. It is important to avoid contact between vaccinated with a live attenuated vaccine or ILT-infected chickens with unvaccinated chickens [5]. Recombinant vector vaccines cannot transmit from bird to bird, and these vaccines are very stable and do not revert to virulence. These vaccines reduce the clinical signs of the disease, but they are not as effective as the live attenuated vaccines in diminishing the shedding of the challenge virus [2].

After infection, the virus replicates in the respiratory and conjunctival mucosa during a 5–12-day incubation period [4]. Excretion of the virus by respiratory and conjunctival secretion occurs as early as two days after infection or four days before the onset of clinical symptoms [11]. During the initial period of replication, the ILT virus is detected in the trigeminal ganglia as early as 2 days after experimental inoculation. Between 2 and 9 days after infection, viral DNA was detected in the conjunctiva and sinuses, followed by the trachea. Viral DNA has also been detected in the ileocecal tonsils and cloaca of experimentally infected chickens [13]. In addition, liver colonization has been described in 20% of 3-day-old chickens, indicating the systematic spread of the virus [14].

The pathological finding shows serous, mucous, bloody or even diphtheria exudate in the mucous membrane of the trachea. Microscopically, lesions are characteristic of alphaherpesvirus infection by the formation of syncytia and intranuclear bodies in the mucosal epithelium [5], which can be detected on the third day after experimental infection. Edema and inflammatory infiltration of cells (lymphocytes, histiocytes and plasma cells) can be observed in the lamina propria of the affected mucosa [14].

The ILT virus has an envelope and is sensitive to heat, ether, chloroform and other lipolytic solvents. Different strains of the ILT virus have different heat resistance [5]. At lower temperatures, the ILT virus maintains infectivity for extended periods. The virus survives for days and months at 13–23 °C in tracheal exudate and chicken carcasses. When stored at −20 °C to −60 °C, the ILT virus survives for months and years. By heating the litter to 38 °C for 24 h and composting it for 5 days, it reduces the virus concentration below the detection level [5]. Chemical disinfectants based on tar, formaldehyde, hypochlorite and iodophor can effectively inactivate the ILT virus [10]. ILT prevention and control measures are based on preventing a contact between viruses and hosts by application of biosecurity measures on farms and vaccination [2,11]. Live attenuated vaccines (derived from tissue culture—TCO or derived from chicken embryos—CEO) and recombinant vaccines (using turkey herpes virus or pox virus as vectors) have been used [2,6,12].

In the territory of Serbia, the last reported case of ILT was almost two decades ago on the farms of broiler breeder flocks [15]. There is no data on the appearance of the ILT virus before this period. At that time, vaccination with a live attenuated vaccine (CEO) was carried out, so all flocks in the epizootic area were vaccinated in the next 2 years. After that period, the vaccination was stopped. No new cases of the disease were reported until the outbreak described in this study.

The aim of our study is to show the data on a new occurrence of the ILT virus in flocks of poultry with respiratory clinical symptoms on farms in Vojvodina, Serbia. Moreover, measures and strategies for the prevention and control of this disease are proposed.

## 2. Materials and Methods

### 2.1. Affected Poultry Flocks

In the period from April to October 2020, respiratory clinical symptoms, unilateral or bilateral swelling of the head, conjunctivitis, increased mortality and decreased egg laying was recorded on 5 farms of laying hens and 1 farm of the broiler breeder flock in Vojvodina (Figure 1). The symptoms were observed in a total of 20 infected poultry flocks, of which 16 flocks of laying hens were in the production phase, 3 flocks of laying hens in rearing period and 1 broiler breeder flock in production phase. The diseased flocks were not vaccinated against ILT before outbreaks.

Data about infected flocks (type of farm, genetic line, age, reared system, number of hens, clinical symptoms, duration of symptoms, characteristically postmortem lesions) are summarized in Table 1.

### 2.2. Collection of Samples

Ten carcasses of dead chickens per flock were sampled from all farms and transported to the Scientific Veterinary Institute “Novi Sad” (Vojvodina, Serbia). A macroscopic examination was performed on all birds. After the pathomorphological diagnosis was completed and the ILT infection was suspected, the organs were sampled for further laboratory tests. For ILT virus testing tracheal tissue from all 10 birds was sampled and a one pooled sample was made per each flock. This means that a total of 20 pooled samples were prepared (Table 2). Detection of ILT virus was performed by real-time PCR. Samples of internal organs (liver, spleen) and swabs of the eye mucosa were examined for bacteriological analysis (Table 2) because it was necessary to examine the presence of coinfections in diseased birds. For histopathological analysis, one bird per flock was collected. We chose samples for histopathology from birds that had characteristic changes for ILT.

Ten birds from each flock were humanely euthanized 28 days after the disappearance of the disease symptoms (after two longest incubation periods, respectively). One pooled sample of tracheal tissue was collected from humanely euthanized birds for molecular testing in order to determine the presence of ILT virus. A total of 20 pooled tissue samples were collected (one from each flock).

### 2.3. Bacteriological Analysis

The conjunctival swabs of the poultry eye were seeded on chocolate agar that provides factor V, necessary for the growth of *Avibacterium paragallinarum* (*Haemophilus paragallinarum*—according to the old nomenclature; *A. paragallinarum*). Chocolate agar was incubated at 37 °C 24–48 h in an incubator with 5% CO2. Suspected colonies of *A. paragallinarum* were sieved from chocolate agar to Columbia agar with the addition of 5% defibrinated sheep blood. Firstly, *A. paragallinarum* colonies were seeded with loop and then secondly, Staphylococcus aureus ATCC25923 was seeded vertically across the lines of seeded *A. paragallinarum* as a feeder bacteria. Moreover, blood agar was incubated at 37 °C 24–48 h in an incubator with 5% CO2. Pure culture of *A. paragallinarum* was identified by gram staining and classical biochemical tests, catalase, oxidase and urease, as well as tests for mobility and carbohydrate fermentation.

Twenty-five grams of liver and spleen were seeded in 225 mL buffered peptone water (CM0509; Oxoid, Basingstoke, Hampshire, United Kingdom) and incubated for 24 h at 37 °C. After incubation, 10 µL of suspension was transferred to McConkey agar per loopful (CM0007; Oxoid). After overnight incubation of McConkey agar at 37 °C, colonies were obtained that morphologically correspond *Escherichia coli* (*E. coli*). Finally, *E. coli* isolates were confirmed by biochemical tests that include oxidase and catalase assays, glucose and lactose fermentation using triple sugar iron agar (TSI), indole production, methyl-red, Voges-Proskauer test and the inability to use citrate as a carbon source (IMViC test).

### 2.4. Molecular Analysis, DNA Extraction, PCR Sample Processing

For the detection of ILTV genome presence, tissue samples of trachea of clinically ill/humanely euthanized (Ethics Committee Opinion no. 05-9821) or found dead animals were chopped with scissors into small pieces in a total amount of 0.2–0.3 g per sample. The prepared samples were placed in 2 mL microtubes and homogenized in 1 mL sterile phosphate-buffered saline for 5 min using a Tissue Lyser LT (Qiagen, Hilden, Germany) operating at 50 Hz. The homogenates were then centrifuged for 10 min at 2000× *g*, and the supernatant was used for DNA extraction. Viral DNA was extracted using the commercial kits Indi Spin Pathogen Kit (Indical Bioscience GmbH, Leipzig, Germany) according to the manufacturer’s instructions. The TaqMan-based real-time PCR was conducted using the commercial kit Quanti Tect Multiplex PCR Kit (Qiagen, Valencia, CA, USA) with the primers and probes that targeted the conserved area of gC gene part of ILTV genome, as described by Callison et al. (2007) [16]. Briefly, the 25 µL PCR reaction mix included 1 µL (800 nM) of each primer (forward: ILTVgCU771 (5′-CCTTGCGTTTGAATTTTTCTGT-3′) and reverse: ILTVgCL873 (5′-TTCGTGGGTTAGAGGTCTGT-3′), 1 µL (200 nM) of dual-labeled fluorogenic probe (ILTVprobe817 (5′-6-FAM-CAGCTCGGTGACCCCATTCTA-BHQ1-3′), 12.5 µL of Quanti Tect Multiplex PCR Master Mix, 4.5 µL of nuclease-free water, and 5 µL of DNA template. The parameters for the real-time assay performed on PCR instrument 7500 Real-Time PCR System (Applied Biosystems, Thermo Fisher Scientific, Waltham, MA, USA) were set as follows: 1 cycle of 50 °C for 2 min, initial denaturation at 95 °C for 10 min, and 40 cycles of 95 °C for 15 sec, and annealing and extension at 60 °C for 1 min.

### 2.5. Histopathology

A total of 20 birds belonging to the different 20 affected flocks from the 6 farms under study, were selected for histopathologic studies (Table 2). Following necropsy, trachea and lungs were examined grossly and collected tissue samples were fixed for 48 h in 10% formalin solution. After the fixation was completed, tissue modeling and tissue incorporation into paraffin were performed. The modeled tissue was first dehydrated through the following series of alcohols: 70%, 96% and absolute alcohol, and then clarified in xylene and impregnated with paraffin. Paraffin molds were cut into 4–5 μm thick sections and mounted on slides. After dewaxing and rehydration, tissue sections were stained by hematoxylin-eosin (HE). Slides were examined under light microscope (Leica Microsystems, LAS EZ).

## 3. Results

### 3.1. Case History

In our study, the first case of ILT infection occurred in April 2020 on a laying hen farm with birds aged 24 weeks (Farm I). The infection occurred in two flocks at the same time. The flocks have not been vaccinated against this disease. At that time, there were no available data on the presence of the virus on the territory of Serbia. Several days after the onset of symptoms in these two flocks, clinical symptoms such as a decrease in egg-laying capacity, a decrease in food consumption and increased mortality appeared in the other 6 flocks. Five days after the occurrence of ILT on the first farm, a suspicion of infection was reported in laying hens in the production and breeding phase on the second farm (Farm II). The age of the flock on this farm was 4, 36 and 57 weeks. The distance between the first and second farms is about 60 km. Clinical symptoms on Farm II appeared 5 days after the entrance of undisinfected and unwashed trucks from the slaughterhouse, which could have been a potential way to bring this viral agent to the farm. One day after the outbreak occurred on Farm II, clinical symptoms were reported on Farm III in the flock in the production. The distance between the second and third farms is about 12 km. Two days after the outbreak occurred on the third farm, the symptoms also occurred on farms IV and V. Farm IV is at the same location as Farm II, while Farm V is at the same location as Farm III. On all five laying hen farms, the hens were kept in cages and it was observed that most hens with developed clinical symptoms were found in the last third of the facility (last cages). Three weeks after the onset of symptoms on Farm V, an ILT infection occurred on Farm VI, in the broiler breeder flock. This farm is about 33 km away from Farm V. It is important to note that the breeder flock on Farm VI underwent interventional vaccination with a live attenuated ILT vaccine (Nobilis ILT, Intervet Pty Ltd.) immediately after the onset of clinical symptoms. The vaccine was administrated via eye drops.

During several months of examination and clinical observation of 20 flocks of poultry from 6 farms infected with the ILT virus, a total of 200 carcasses of dead birds and 200 carcasses of humanely euthanized birds were examined—40 pooled samples using molecular diagnostics (20 pooled samples originating from diseased flocks at the time of infection and 20 pooled samples 28 days after the absence of clinical signs), and 20 pooled organ samples (liver and spleen) and 20 swabs by bacteriological analysis (Table 2). The data on infected flocks, course of infection and characteristics of the epizootic are shown in Table 1.

### 3.2. Clinical Manifestation

All the affected flocks in this study were clinically examined. The clinical examination of the flock revealed mild to severe respiratory symptoms including, the swelling eyelids, infraorbital sinuses and adjacent areas, severe nasal discharge, conjunctivitis, increased tearing, panting and stretching of the head and neck with an open beak (Figure 2). A decrease in feed consumption (up to 40.0%), weight loss, a decrease in egg production (up to 42.0%) and increased mortality (up to 31.5%) were also reported.

### 3.3. Gross Findings

At the postmortem examination, remarkable findings were localized in the upper respiratory tract. Inflammation of the laryngeal and tracheal mucosa, the presence of mucous exudate and yellow caseous plaques in the trachea and larynx were observed as the dominant findings, while the tracheal lumen was obliterated (partially or totally) (Figure 3a–d). Common findings were hemorrhagic tracheitis with hemorrhagic mucus along the entire trachea (Figure 3b). The tracheal mucosa was highly congestive and cyanotic. In several cases, congested lungs were reported.

### 3.4. Histopathology

Microscopic examination of the trachea revealed inflammation with lymphocytic infiltration. The tracheal lumen is filled with exudate containing fibrin, inflammatory cells (lymphocytes) and syncytial cells (Figure 4a). In most cases, eosinophilic intranuclear inclusion bodies were found in the laryngeal epithelium (Figure 4b). Epithelial cells are necrotic and desquamated (Figure 4c). Edemic fluid in the lung tissue was present (Figure 4d).

### 3.5. Molecular and Bacteriological Findings

The presence of ILT virus was confirmed by PCR in tracheal tissue samples originating from dead individuals with clinical symptoms and established changes on pathomorphological examination. A positive finding was obtained in all flocks (Table 3).

The cessation of clinical symptoms was observed 2–4 weeks after the first reported case in the flock. A repeated analysis for the presence of the ILT virus using the PCR method was performed 28 days after the last reported clinical case of ILT in the flock. The result was negative in all flocks (Table 3).

Bacteriological analysis of internal organs and swabs of the eye mucosa originating from individuals with a clinical picture showed the presence of *E. coli* predominating in most individuals and *A. paragallinarum* in several flocks (Table 3).

The clinical signs of the infected birds indicated the suspicion of infectious coryza; therefore, an examination was conducted for the presence of coinfection with *A. paragallinarum*.

## 4. Discussion

ILT is a contagious disease of poultry and is spread all over the world. It is present in regions with developed poultry farming, and it most commonly appears on industrial farms and also in poultry raised in extensive conditions [4,17]. The last time this disease was recorded on the territory of the Republic of Serbia was almost two decades ago when Orlić and associates (2003) recorded it in three broiler breeder flocks [15]. In April 2020, the ILT infection appeared on the laying hen farm, and after that, it spread to four more laying hen farms and one farm of broiler breeders. The diseased flocks were not vaccinated against ILT. In the countries of our region, outbreaks of this disease have been reported in two flocks of commercial layers in North Macedonia [18], in commercial and backyard poultry flocks in Slovenia [17] and in an organic broiler farm and surrounding flocks in Greece [19].

The disease spread slowly in the flock horizontally through the contact of sick hens with healthy ones [10]. In addition, the virus is known to spread through farm equipment, vehicles, bedding, dust, drinking water, insects, crows, cats and dogs [4,5]. In our study, it is likely that undisinfected and unwashed slaughterhouses and rendering plant vehicles were a potential breach of biosecurity, leading to health risks for birds. If biosecurity measures on the farm are insufficient, there is a greater risk of ILT.

The disease in the affected flocks is detected based on the clinical signs and macroscopic findings, and the final diagnosis is confirmed by the molecular PCR method. Following the confirmation of the ILT virus, measures to prevent the spread and eradication of the virus in infected and endangered areas are prescribed by the competent veterinary institute are applied. The measures were similar to the ones that are prescribed to prevent the spread of other infectious diseases (avian influenza, Newcastle disease, etc.). The recommended measures consisted of raising the level of biosecurity measures on the farm to a high level and daily informing the epidemiological service of the institute about the health condition of the poultry on the farm.

ILT virus infection is characterized by respiratory symptoms, and the first clinical symptoms include dyspnea, tearing of the eyes, swelling of the head and neck with an open beak [6,20]. These symptoms were also observed in infected flocks in our study. According to Garcia et al. (2013) and Wolfrum (2020), more severe clinical symptoms appear in the form of panting, dyspnea and expectoration of bloody mucus [3,8]. Hens with a more severe clinical picture in our study had symptoms such as panting and stretching of the head and neck with an open beak, the so-called “hunger for air”, while coughing up blood was not recorded. Coughing up blood occurs in acute cases and depends on the virulence of the virus [3,7]. The clinical picture of the disease in many cases does not include the acute signs (nasal discharge, lacrimation, moist rales, expectoration of blood, etc.), which may be a feature of a virus of low pathogenicity [17] or reflect immunity stimulation by vaccination. In our case, the disease was first noticed in the last third of the building, in the last cages. As the disease progressed, the virus spread to the first two-thirds of the facility. We did not detect this kind of phenomenon of spreading the ILT virus in the facility, as described in the available literature [4,5]. In addition to milder and more severe respiratory symptoms, unilateral or bilateral swelling of the head, nasal discharge, inflammation of the conjunctiva of the eye, increased tearing, closed one or both eyes were observed. These symptoms have also been described by other authors [7,10,21]. With the development of the disease, there was a loss of body weight due to the reduction of food and water consumption, which is a very common accompanying symptom [12].

With the further development of the disease, increased mortality appeared (0.8–31.5%) and the number of dead individuals largely depended on the condition of the flock, the age of the flock and the period of the production cycle. Up to 9.2% of deaths occurred in breeding flocks, up to 5% in the flocks at the beginning of the production cycle, and up to 31.5% in older flocks in the last phase of the production cycle (Table 1). According to the data of Aras et al. (2018), the percentage of deaths in laying hens infected with the ILT virus was 7.93% and according to Tamilmaran et al. (2020), up to 17%, while Molini et al. (2019) reported mortality rates of up to 70% [22,23,24]. This shows that the percentage of dead hens is variable and depends on several factors, and the same findings/conclusions were reported in other studies. The results from the study by Kirckpatric et al. (2006) revealed that ILTV strains vary considerably in their capacity to induce mortality, clinical signs and lesions in different tissues [25].

In our study, the increase in mortality was accompanied by a decrease in egg laying; it fell by 2.7–42.0%, depending on the age of the flock and the condition. Most authors describe the clinical picture with respiratory symptoms, high morbidity and variable mortality, with a significant decrease in egg laying [8,15,24]. In our research, the egg-laying capacity of flocks fell by 15% at the beginning of the production period, while egg laying fell by 42% in the middle of the production cycle. Egg laying in the hens that were in the last phase of the egg production cycle fell to 35%. In the broiler breeder flock, the decrease in egg laying was 2.7%, but in this flock, the mortality rate was also lower (0.8%), as well as the duration of the disease. According to the results from a study previously published of Orlić et al. (2003), the percentage of mortality in infected breeder flocks was 16%, while the decrease in egg-laying capacity was 30% [15], which was a higher percentage compared to the data obtained in our study. We assumed that such a small percentage of egg laying, low mortality and shorter duration of symptoms on Farm VI were the result of vaccination that prevented the spread of field virus in the flock from infected to healthy individuals and prevented replication of field virus in the trachea. This phenomenon has been described by other authors as well [2,26].

Macroscopically, changes in birds infected with the ILT virus are most pronounced in the upper respiratory tract, tracheal and laryngeal mucosa. ILT is characterized by the presence of mucosal exudate and yellow caseous plaques in the trachea and larynx, inflammation of the mucosa, hemorrhagic tracheitis with blood clots and hemorrhagic mucus along the entire trachea [27,28,29]. Very often, the mucous membrane of the trachea is very congested and cyanotic, and at the entrance to the larynx, there is a caseous yellowish clot that further fills the lumen of the larynx and trachea [23,30], which was found in our study. Previously published findings in ILT cases describing sinusitis and conjunctivitis associated with almond-shaped eyes [5,10] were similar to the ones noted in the outbreaks under our study. Bacteriological analysis of internal organs and swabs of the eye mucosa in most hens revealed the presence of *A. paragallinarum*, which indicates a synergistic effect between the ILT virus and this bacterium [31]. In our study, flocks infected with ILT and *A. paragallinarum* had a mortality between 0.8 and 31.5%, decreased egg laying by 2.7–31.6% and decreased feed consumption by 2.1–5.5%. According to the data of Jonare et al. (2020), coinfection with *A. paragallinarum* was the most frequent in flocks infected with ILT [32]. Couto et al. (2016) described that mixed infections of ILT with *A. paragallinarum* can result in a shorter period of incubation and increased mortality [33]. Moreover, they report that mixed infections of ILT with *E. coli* and *A. paragallinarum* are common [33], which is in accordance with our results. Our results showed that the highest mortality (31.5%) was in the flock that was the oldest and had coinfection with *A. paragallinarum*, while the highest decrease in egg laying (42.0%) and decrease in feed consumption (40.0%) were in the flocks that did not have coinfection with *A. paragallinarum* (Table 1). Coinfections can cause increased mortality, as shown by the results of Tamilmaran et al. (2020) [23] and Zorman Rojs et al. (2021) [17].

Microscopically, the occurrence of a syncytium with eosinophilic intranuclear inclusion bodies is considered a pathognomonic finding in ILT virus infection. Herpesvirus inclusion bodies are clusters of viral particles, proteins and genomes and most commonly occur in severe and moderate forms of the disease [20]. Our study showed the same result, which confirmed the diagnosis by histopathology. Histopathological diagnosis is considered a fast and reliable method for confirmation of ILT in the acute phase of the disease (OIE 2009) [34], but the chance of detecting typical lesions is drastically reduced if the method is performed in subacute and chronic disease (8–10 days after infection), due to epithelial desquamation cells [35]. In the cases where typical pathomorphological lesions are missing, the use of the molecular PCR method is recommended for the final diagnosis of the disease. The PCR method is a much more sensitive method compared to virus isolation, and the real-time PCR method has been successfully applied in this research to establish a final diagnosis of the disease [20]. Strategies for controlling ILT are usually based on preventing a contact between the virus and the host through biosecurity measures or vaccination [2,11]. Shortly after May and Tittsler first described ILT in 1925, chicken immunization by inoculation of the virulent virus through the cloaca began. This is considered to be the first effective vaccine developed for a serious viral disease of poultry [6]. Attenuated live vaccines were then developed by successive passages of the virulent virus in cell culture (derived from tissue culture—TCO) or in embryonic eggs (derived from chicken embryos—CEO) [2]. These vaccines are now used in commercial poultry flocks around the world. In recent years, recombinant vaccines have been produced using turkey herpes virus (HVT) or fowlpox virus (FPV) as a carrier for ILT glycoproteins, which can elicit a protective immune response in vaccinated chickens [6,12]. These vector vaccines are characterized by the impossibility of transmission from bird to bird and are very stable, so they cannot restore virulence [2,12]. Applying good biosecurity measures on farms can prevent ILT. Using a geographic information system (GIS) that provides information from the region on biosecurity action plans, quarantines, vaccinations and ILT outbreak sites or slaughterhouses, a strategy can be developed to control and combat ILT [10,11]. Following the introduction of recombinant vaccines into vaccination programs, there has been a need to use a combined vaccination strategy in the areas with a highly virulent ILT virus. This strategy includes the administration of a recombinant vaccine in the hatchery and vaccination with attenuated live vaccines during the rearing period in order to improve immune protection [2]. Immune protection is achieved in a week by using ILT live attenuated vaccines. This period is 4 weeks for recombinant vaccines. It is important to note that humoral immunity is not the main immune response to the ILT virus in chickens. Research has confirmed that the role of cell-mediated immunity in the fight against the ILT virus is significant [5]. Local cell-mediated immune responses in the trachea are the principal mechanism in defense against the ILT virus [3]. The results from a study by Maekawa et al. (2019) show that prior vaccination with rHVT-LT reduced the CEO vaccine replication and suggest that the combination of rHVT-LT + CEO vaccines in the long term may reduce the circulation of CEO viruses [2]. Therefore, this combined vaccination strategy offers a safer alternative than the uninterrupted use of solely CEO vaccines in poultry facilities. Future strategies for preventing the occurrence of ILT on farms in Vojvodina will include the procurement of flocks vaccinated with recombinant vaccines on the first day of incubator life and the use of live attenuated vaccines during the rearing period.

## 5. Conclusions

Infectious laryngotracheitis of poultry is a very important disease for modern poultry, and it is on the list of diseases that must be reported. After almost two decades, it appeared in Vojvodina on a broiler breeder flock farm and laying hen farms. Based on the characteristic clinical picture and pathoanatomical findings, ILT was suspected and the presence of the ILT virus was confirmed using the molecular PCR technique. Morbidity was up to 45%, and mortality was up to 31.5%. The decline in egg production was more than 40%, but in 2–4 weeks it returned to the level of technological norms. Bacteriological analysis in several flocks confirmed the presence of *A. paragallinarum*. In our study, it is suspected that the virus entered two farms through the unwashed and disinfected slaughterhouses and rendering vehicles. This shows that the application of strict biosecurity measures on farms can prevent the occurrence of ILT. Future strategies for the control and prevention of infectious laryngotracheitis will include the procurement of flocks vaccinated with recombinant vaccines or the registration of live attenuated vaccines and their use during the rearing period.

## Figures and Tables

**Figure 1 animals-12-03551-f001:**
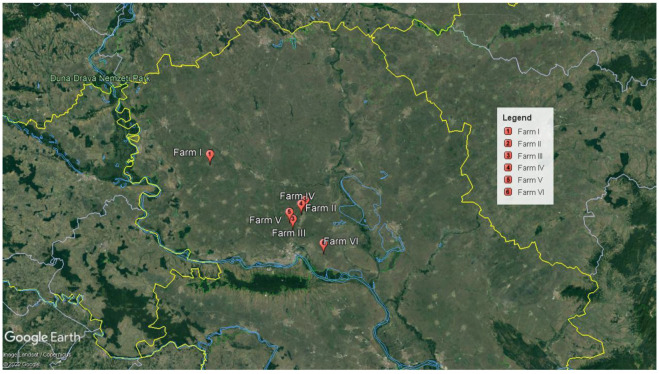
Map of Vojvodina province and locations of the affected farms. Distance from Farm I to Farm II was 46.00 km; from Farm II to Farm III was 9.46 km; from Farm III to Farm IV was 7.38 km; from Farm IV to Farm V was 6.17 km; from Farm V to Farm VI was 19.92 km. The sequence of occurrence of ILT on the farms follows the numerical designation of the farm (Farm I is the first affected and Farm 6 is the last affected). Estimated poultry population in this area is as follows: layer hens 1,481,000; broiler breeders 260,000 and broilers 1,200,000.

**Figure 2 animals-12-03551-f002:**
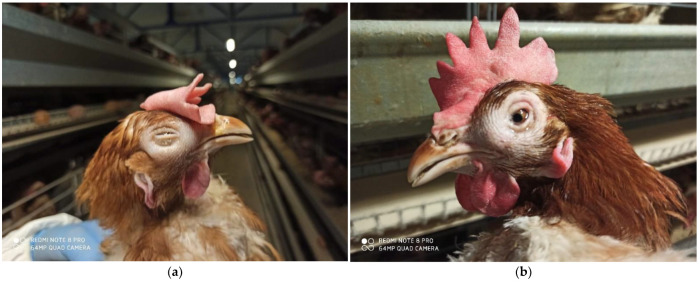
(**a**,**b**) Diseased hens with clinical symptoms—bilateral swelling of the head, runny nose, conjunctivitis, increased tearing.

**Figure 3 animals-12-03551-f003:**
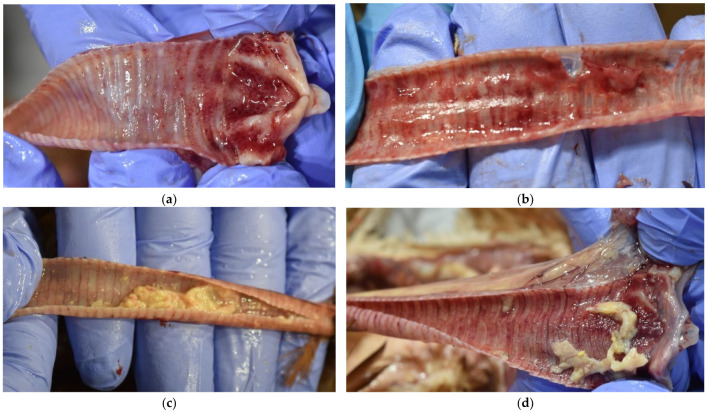
(**a**,**b**) Inflammation of the mucous membranes of the larynx and trachea—hemorrhagic exudate; (**c**,**d**) yellow caseous plaques in the lumen of the trachea.

**Figure 4 animals-12-03551-f004:**
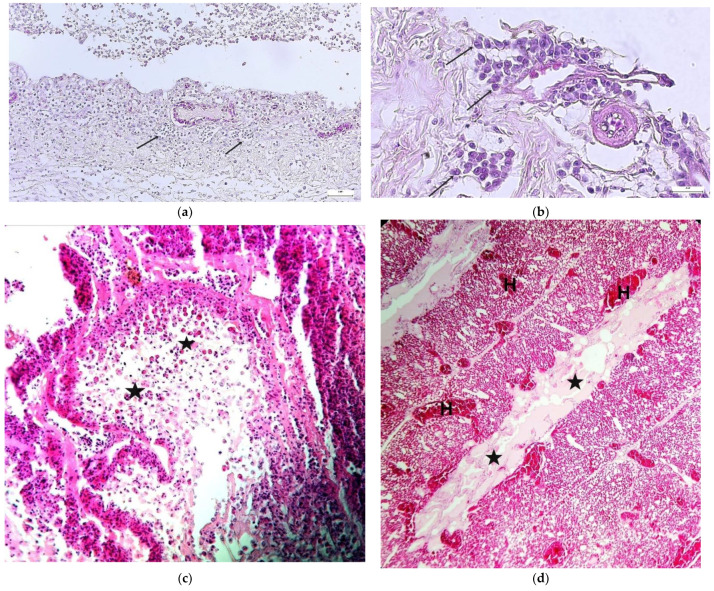
(**a**) Fibrinous exudate in the trachea with lymphocytic infiltration (arrows) in mucosa, H&E stain, magnification × 200; (**b**) larynx, large multinucleated syncytial cell showing many eosinophilic intranuclear inclusion bodies (arrows), H&E stain, magnification × 600; (**c**) Cross section of the trachea showing sloughed epithelial cells (stars), H&E stain, magnification × 200; (**d**) lungs. Hyperemia in blood vessels (H) and endemic fluid in the lung tissue (stars), H&E stain, magnification × 100.

**Table 1 animals-12-03551-t001:** Data on diseased chicken flocks from all farms and data on clinical symptoms.

Farm Number	Flock Type	Flock Age (Weeks)	Genetic Line	Reared System	Number of Hens in a Flock	Clinical Symptoms	Morbidity (%)	Mortality (%)	Egg Laying Decrease (%)	Feed Consumption Decrease (%)	Duration of Symptoms (weeks)	Characteristic PM Lesions
I	LH *	26	Lohmann brown	cage	23,850	+	35.0	3.7	12.0	30.7	3	+
	LH *	25	Lohmann brown	cage	23,136	+	40.0	3.2	14.0	27.2	3	+
	LH *	58	Lohmann brown	cage	23,341	+	45.0	6.4	15.0	28.9	3	+
	LH *	59	Lohmann brown	cage	22,910	+	45.0	6.6	14.0	29.1	2	+
	LH *	61	Lohmann brown	cage	22,280	+	40.0	5.1	15.0	22.8	3	+
	LH *	70	Lohmann brown	cage	22,621	+	45.0	6.6	16.0	33.3	2	+
	LH *	66	Lohmann brown	cage	18,828	+	30.0	2.6	18.0	27.2	2	+
	LH *	21	Lohmann brown	cage	22,970	+	40.0	6.0	16.0	29.8	3	+
II	LHR	4	Lohmann brown	cage	33,895	+	40.0	8.8	/	15.8	3	+
	LHR	4	Lohmann brown	cage	32,868	+	40.0	8.6	/	16.2	3	+
	LH *	57	Lohmann brown	cage	31,911	+	25.0	4.5	20.5	5.2	3	+
	LH *	57	Lohmann brown	cage	31,531	+	35.0	8.5	31.6	5.5	3	+
	LH *	36	Lohmann brown	cage	33,075	+	20.0	3.1	18.8	4.0	3	+
	LH *	36	Lohmann brown	cage	23,504	+	20.0	3.1	28.9	4.7	2	+
III	LH *	59	Lohmann brown	cage	43,500	+	35.0	6.7	23.4	3.4	4	+
	LH *	111	Lohmann brown	cage	24,700	+	45.0	31.5	26.7	3.0	3	+
IV	LH *	58	Tetra SL	cage	3750	+	30.0	6.5	35.4	40.0	2	+
V	LHR	12	Lohman brown	cage	15,300	+	40.0	9.2	/	5.6	3	+
	LH *	42	Lohman brown	cage	14,600	+	35.0	8.9	42.0	12.3	4	+
VI	BBF *¤	49	Ross 308	floor	16,120	+	15.0	0.8	2.7	2.1	2	+

LH—laying hen flock in the production phase; LHR—laying hen flock in rearing phase; BBF—broiler breeder flock; PM—postmortem lesions for ILT; + with symptoms; − no symptoms; * During the rearing phase, the hens were vaccinated against Marek’s disease, *Salmonella* Enteritidis and *Salmonella* Typhimurium, Gumboro disease, infectious bronchitis, *Newcastle* disease, fowlpox, egg-drop syndrome; After the onset of symptoms, the flock was vaccinated with live attenuated ILT vaccine; All flocks were not vaccinated with live vaccination for other respiratory viruses during the last 2 weeks before sampling; Data about mortality, egg lay decreasing and feed consumption decreasing show the information when the birds were sick due to ILT.

**Table 2 animals-12-03551-t002:** Data on the number of collected samples from the examined flocks.

Farm Number	Flock Type	Flock Age (Weeks)	Number of Samples for PCR (Pooled Samples)		Number of Samples for Bacteriology (Pooled Samples)		Number of Samples for Histopathology
			Trachea (Diseased Birds)	Trachea (Birds without Clinical Signs) *	Organs (Liver, Spleen)	Tracheal Swabs	Organs (Trachea, Lung)
I	LH	26	1	1	1	1	1
	LH	25	1	1	1	1	1
	LH	58	1	1	1	1	1
	LH	59	1	1	1	1	1
	LH	61	1	1	1	1	1
	LH	70	1	1	1	1	1
	LH	66	1	1	1	1	1
	LH	21	1	1	1	1	1
II	LHR	4	1	1	1	1	1
	LHR	4	1	1	1	1	1
	LH	57	1	1	1	1	1
	LH	57	1	1	1	1	1
	LH	36	1	1	1	1	1
	LH	36	1	1	1	1	1
III	LH	59	1	1	1	1	1
	LH	111	1	1	1	1	1
IV	LH	58	1	1	1	1	1
V	LHR	12	1	1	1	1	1
	LH	42	1	1	1	1	1
VI	BBF	49	1	1	1	1	1
Total			20	20	20	20	20

LH—laying hen flock in the production phase; LHR—laying hen flock in rearing phase; BBF—broiler breeder flock; * 28 days after the symptoms were not present anymore.

**Table 3 animals-12-03551-t003:** Results of bacteriological and molecular analyzes from all farms.

Farm Number	Flock Type	Flock Age (Weeks)	ILTV qPCR *	Bacteriology	ILTV qPCR **	Farm Number	Flock Type	Flock Age (Weeks)	ILTV qPCR *	Bacteriology	ILTV qPCR **
I	LH	26	+	*E. coli*	-	III	LH	59	+	*E. coli, A. paragallinarum*	-
	LH	25	+	*E. coli*	-						
	LH	58	+	*E. coli*	-						
	LH	59	+	*E. coli*	-						
	LH	61	+	*E. coli*	-						
	LH	70	+	*E. coli*	-						
	LH	66	+	*E. coli*	-		LH	111	+	*E. coli, A. paragallinarum*	-
	LH	21	+	*E. coli*	-						
II	LHB	4	+	*E. coli*	-	IV	LH	58	+	*E. coli*	-
	LHB	4	+	*-*	-	V	LHB	12	+	*-*	-
	LH	57	+	*E. coli, A. paragallinarum*	-		LH	42	+	*E. coli*	-
	LH	57	+	*E. coli, A. paragallinarum*	-	VI	BBF	49	+	*E. coli, A. paragallinarum*	-
	LH	36	+	*E. coli*	-						
	LH	36	+	*E. coli*	-						

LH—laying hen flock in the production phase; LHB—laying hen flock in breeding phase; BPF—broiler breeder flock; * real-time PCR (qPCR) analysis of ILT virus at the time of clinical symptoms; ** real-time PCR (qPCR) analysis of ILT virus 28 days after the last observed clinical case in the flock; + positive finding; − negative finding; flocks previously examined for *Mycoplasma synoviae* during the production cycle and were serologically positive; All flocks were examined for *Mycoplasma gallisepticum* earlier in the production period and were serologically negative; ¤ antibiotic therapy with oxytetracycline was implemented in all flocks infected with *A. paragallinarum*.

## Data Availability

Not applicable.

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
