# Peer review of "Diagnosis of Infectious Laryngotracheitis Outbreaks on Layer Hen and Broiler Breeder Farms in Vojvodina, Serbia"

_animals, 2022, doi:10.3390/ani12243551_

Round 1

Reviewer 1 Report

Comments to the author

Occurrence of Infectious Laryngotracheitis Outbreaks on Poultry Farms and Prevention Strategies by Pagic et al.

The objective of this study was to document an outbreak of infectious laryngotracheitis (ILT) in laying hens and broiler breeders in a specific region (Vojvodina). In this article, the author reports the clinical findings and the tool used to confirm the ILT infection. Also, it was detected coinfection of ILT with Avibacterium paragallinarum in some flocks.

Major issues:

The major issue with this case report is the use of pools samples for PCR and bacteriology diagnostic. It would have been important to know the individual chickens results to properly determined the evidence of coinfection and try to correlate the PCR results with the histopathological findings. 

Specific comments: 

Line 14: Include the information of the region (Vojvodina).

Line 18-19: To reach that conclusion, you would need evidence that in new vaccinated flocks you did not see outbreaks of the diseases. Line 32 indicate that you will implement a future control and strategies.

Line 21: ILT is caused by a Gallid alpha herpesvirus-1 (GaHV-1).

Line 25: Eliminate the phrase “air hunger”

Line 28: “upper parts of respiratory tract” change to “upper respiratory tract”

Line 34: In the abstract section, it would be important to mention that some flock presented coinfections of ILT and Avibacterium paragallinarum. 

Line 39: ILT is caused by a Gallid alpha herpesvirus-1 (GaHV-1).

Line 42: Eliminate “or”

Line 43: Eliminate “almond-shaped eyes”

Line 46: “Shortness of breath” change to “severe dyspnea”

Line 47: What do you mean “attack the peripheral nerves” Maybe establish latency in the trigeminal ganglia.

Line 48: “start of egg laying” change to “peak of egg production”

Line 49: “secretion” change to “excretion”

Line 50: Eliminate the word “intensive”

Line 52: Eliminate “susceptible flocks”

Line 55: Find a better word than “individuals”

Line 56: Replace “survive” by “persist”

Line 59 – 61: Clarify the idea. Once the Alphitobius are infected, they transmit the virus until 42 days after infection?

Line 62 – 64: You should point “vaccinated with live attenuated vaccines”. 

Line 65: According literature, ILT is only reactivated under stress. There is no spontaneous reactivation. 

Line 65: Find a better word for “overpopulation”. Maybe high stocking density?

Line 65 – 67: Change the word “moving” by “being transmitted”

Line 67 – 68: The aforementioned sentence does not support ILT vaccines can only be used in endemic areas. Are you talking about CEO vaccines? What about recombinant ILT vaccines? Recombinant ILT vaccines do not transmit horizontally and do not regain virulence. 

Line 68 – 69: Vaccinated chickens with which vaccine? CEO vaccines? Recombinant ILT vaccines do not spread horizontally. 

Line 70: Change the word “multiply” by “replicate”

Line 71: Replication and incubation period are two different things. ILT virus can stop replicating but you may still be able to see some clinical signs. 

Line 73 – 75: What about ILT detection in the trachea and conjunctiva?

Line 79 – 80: There is no report of bloody or diphteric exudate in the conjunctiva. The lesions that you refer are for the trachea. 

Line 80 – 83: In the sentence “lesions are characteristic of herpesvirus infection” You need to clarify what herpesvirus are you referring. For example, Marek’s disease is caused by a Herpesvirus but do not produce those lesions. 

Line 83 – 84: What type of inflammatory infiltration? Lymphocytes? Macrophages?

Line 86: It is important to cite the study that prove “Different strains of ILT virus have different heat resistance” 

Line 87: What do you mean “The virus is ineffective for extensive periods”. Ineffective inactivated? Ineffective infectious? Please clarify. 

Line 88: Stored what? 

Line 89: Replace the word “bedding” by “litter”

Line 90 – 91: What do you mean by detection level? Detection level by PCR?

Line 100: “attenuated vaccines” change to “Live attenuated vaccines”. What attenuated vaccine TCO or CEO?

Line 110: Did you observe conjunctivitis” If so, please add that information.

Line 116-146: There is an overlap in information between these sentences and the table 1. I suggest to stick only with the table. You can add into the table the information  about genetic lines (lohman), system (cage). The information about how flocks possible got infected are already described in the case history (line 245). 

Figure 1: Can you describe:

What is the distance between the farms?

What was the sequence of infection. What farm was affected first and so on.

What is estimated chicken population in that area?

Table 1: Can you clarify the mortality? Does the data show the accumulative mortality? The weekly mortality? The mortality due to ILT?

What is the period of egg lay decreasing? One week after infection? Or the egg reduction showed after birds finished production. 

Does the feed consumption reduction show the information when the birds were sick?

Line 161: Change the word “individual” to “chickens”

Line 164: change the word “virus” to “infection”

Line 191: During infection, did you implement antibiotic treatment. If so, please add that information. 

Line 214 – 214: How were chickens euthanized? Cervical dislocation? CO2?

Line 282: Eliminate the term “hunger for air”.

Line 293 – 294: Eliminate “Bleeding on the laryngeal and tracheal mucosa” “blood clots”

Line 297: Add the information in parenthesis “Figures are not shown”.

Line 301 – 302: Can you define the inflammatory cellular infiltrate? Lymphocytes? Macrophages? 

Figure 4: The current figures did not allow to see the characteristic microscopic lesions of ILT such as the syncytial cell formation and intranuclear inclusion bodies. Please replace these figures with other ones that show the detail (Greater magnification).

Line 361: Replace “shortness of breath” to “Dyspnea”

Line 362: Eliminate “stretching of the head”

Line 366: Can you discuss why you did not find chickens expectorating bloody mucous. Is this finding ILT strain related?

Line 381: Can you speculate why older flock presented the highest mortality? Stocking density? Stress? Heavy weight?

Line 413 – 415: Can you discuss in a greater extent the impact of Avibacterium paragallinarum? Do the flocks with coinfection show greater mortality and performance affection? What was particular in the flocks with coinfection compared to flock with solely ILT. 

Line 449: Immune protection for live attenuated vaccines takes at least one week. For recombinant vaccines, to get the most of this vaccine it may take until 4 weeks post vaccination. 

Line 450: Why is relevant to mention that ILT protection is not achieved by humoral immunity? That is why inactivated vaccines are not commonly used. 

Author Response

Please see the attchment.

Reviewer 2 Report

Dear authors

The authors provided a manuscript describing a case series of outbreaks of a widely known disease as infectious laryngotracheitis in laying hen and broiler breeder farms from April 2020 to August 2021 in Vojvodina, Serbia. The manuscript described in detail the clinical signs, gross, molecular, and histopathologic confirming the viral etiology and bacteriologic studies describing concomitant bacterial infections in studied birds. The authors must include a permit number/code for proper handling and euthanizing birds from a local ethics committee for animal welfare. Because the authors highlighted the presence of this viral disease after 20 years in Serbia, it could use not only to follow the below-listed suggestions and changes but also to add a mini-review of the few previously reported outbreaks of this entity in the mentioned country and/or neighboring countries belonged to the former Yugoslavia. Several minor edits are suggested, as follows:

- L2-3 The title could be modified. An option could be "Diagnosis of infectious laryngotracheitis outbreaks on layer hen and broiler breeder farms in Vojvodina, Serbia"

L13 Please add a comma after "In our study."

L24 Please delete colon after "were".

L25 Please delete "("air hunger")"

L28 Please delete "parts of"

L31 Please change from "3-4 weeks..." to "Three-to-four weeks..."

L43 Please delete "(almond-shaped)".

L47-48 Please consider the infectious laryngotracheitis virus can be latently located only in the trigeminal ganglia. Please rephrase "... the ILT virus attacks the peripheral nerves and latent infection occurs."

L81-82 Please include the term "syncytia" instead of "multinucleated (multinuclear) cells". Syncytia are composed of multiple eosinophilic intranuclear inclusion bodies.

L104-106 Please begin the paragraph with "The aim of our study...", it could be useful to add "Serbia" after "Vojvodina".

L115 It could be useful to add a brief sentence to introduce the information about each of the farms under study.

L117 Please simply this sentence. Please consider using "...26 weeks, housing 23,000 hens" instead of "...26 weeks, and there were approximately 23.000 hens in them"

L145, L161 Please replace the term "individuals" by using "birds", for example.

L147-148 This sentence could be deleted because the legend of Fig. 1 is in line 150.

L151 The heading of Table 1 could be improved. For example, the term "poultry" is too general and could be replaced by "chicken"

L156-157 Please replace "smallpox and diphtheria" by using "fowlpox".

L157-158 Please delete the symbols before "After" (L157) and "All" (L158).

L162 Please add the location (Vojvodina, Serbia) after naming the Institute.

L162-163 The term "pathomorphological" could be replaced by other ones such as "gross", "postmortem" or "macroscopic".

L163 Please change from "...was performed in the necropsy room" to "...was performed in all birds".

L170-171 Please rewrite this sentence to explain that sampling of one bird per affected flock was included in the histopathologic study.

L173-176 Please rewrite this sentence.

L177 Please consider Table 2 is not completely necessary in the manuscript because no valuable information is added.

L180-191 Please relocate this paragraph to the Discussion or Conclusion sections.

L193 to 201 Please clarify and simplify the culture of conjunctival swabs from affected chickens and Avibacterium paragallinarum-suspected colonies in chocolate, Columbia, and blood agar plates including temperatures, time of incubation, and atmosphere.

L201-203 This paragraph could be relocated to the Results section.

L204 Please use words to replace "25" at the beginning of the sentence.

L207 to 212 This paragraph could be relocated to the Results section.

L207 Please delete "Basingstoke, Hampshire, United Kingdom" because it was above included in the manuscript.

L215 it is mentioned that some of the analyzed samples came from euthanized birds. Thus, the authors must include the term "humanely euthanized" and the permit number/code from a local ethics committee for animal welfare to handle and euthanize birds properly, 

L235-236 Please use the sentence " A total of 20 birds belonging to the different 20 affected flocks from the 6 farms under study, were selected for histopathologic studies".

L246 Please begin the paragraph with "In our study,..."

L250-251 Please delete parenthesis before "decrease" and after "mortality". Please use "such as" or similar.

L257 If you are referring to the infectious laryngotracheitis virus, please use use "this viral agent", "the studied viral agent" or similar. 

L267-268 The mentioned features of this vaccine could be deleted because did not provide any valuable information to the manuscript.

L269 Please use the sentence "The vaccine was administrated by eye drop"

L271-272 The term "individuals" could be replaced by "birds" or a similar one. When you referred to "200 carcasses of sacrificed..." please use the term "humanely euthanized" instead of "sacrificed".

L274 Please use "...28 days after the absence of clinical signs" instead of "...28 days after the symptoms were not present anymore".

L279 Please clarify that the clinical examination was performed in each of the affected flocks included in this study. 

L280 to 282 Please use "swelling eyelids, infraorbital sinuses, and adjacent areas", "severe nasal discharge (including the type of exudate)", and delete "the so-called "hunger for air".

L282 to 284 Please include the percentages of each of the mentioned parameters to know how severe were.

L289 to 297 There are several changes or suggestions for this paragraph        - Please use "...remarkable findings were localized in the upper respiratory tract." instead of "...the changes were localized in the upper parts of the respiratory tract."                                                                                                - Regarding the gross findings, "In pathomorphological examination," could be replaced by " Macroscopically,", " At the postmortem examination," "At necropsy,". These last three last terms are widely used.                                    - L291 The term "diphteroid deposits" could be replaced by "caseous pseudomembranes" or "caseous plaques"                                                        - L292 When you refer to a closed trachea, the proper term to characterize it such as in this case is "obliterated (partially or totally)".                                    - L293 "What do you mean by "bleeding"? Are you describing the presence of blood clot/s?                                                                                                 - L295 In the case of ILT, the tracheal mucosa is usually highly congestive. but not cyanotic. Please use past tense to describe this finding.                      - L296 Please consider using "congested lungs". Sinusitis and conjunctivitis were already mentioned in the clinical signs.

L298-299 The term "diphtheroid deposits" could be replaced as above mentioned.

L301-302 The sentence beginning "Microscopic examination..." could be deleted.

L303 Where are locarted the eosinophilic intranuclear inclusion bodies? Please consider that inside the syncytia surrounded by exudate, sloughed, and necrotic tracheal epithelium.  What type of inflammatory cells are you referring to?

L304 What type of "damage" are you referring to? Necrosis?

L307 to 311 Please find below the suggestions for this paragraph                    - L307 Are you describing "fibrin exudate"? Can you describe the type of inflammatory cell infiltration you are seen?                                                      - L308 Please include the term "eosinophilic" to refer to the intranuclear inclusion bodies. Are they forming syncytia?                                                     - L309 When you described "desquamated cells", it could be useful to include other terms also such as "necrotic", "sloughed", "filling the tracheal lumen", "surrounded by fibrin exudate", etc.

L322-323 Please use the abbreviated names of the isolated bacteria.

L323 to 325 Please rewrite this small paragraph to clarify it.

L342-343 Please delete "A similar way of spreading is described by Parra et al. (2016) [10]". Please use "[10]" after "healthy ones" in L342.

L345 Please use past tense after "plant vehicles".

L346 Please change from "...a potential cause of infection..." to "a potential breach of biosecurity leading to health risk for birds"

L348 Please use past tense and replace "pathomorphological" as above mentioned.

L354 to 360 The information included here is valuable, but the paragraph is too long. Please shortened it. 

L372 Are you describing nasal discharge, oculonasal discharges, or ocular discharges?

L389 Please begin the sentence with "In our study,"

L397 Please clarify that the study of "Orlic et al. (2003)" was "previously published"

L399 Please use past tense for "is" and "assume".

L404 Please replace "pathomorphological" and "individuals" are above suggested.

L406 Please avoid using "diphtheroid". This caseous exudate can be seen in the mucosa of the obliterated trachea and larynx.

L407 Please the term "mucositis" is wrong. Please replace it with a proper one.

L411 to 413 An option to replace this sentence is "Previously published findings in ILT cases describing sinusitis and conjunctivitis associated with almond-shaped eyes [5,10] were similar to the ones noted in the outbreaks under our study"

L414-415 Please include the abbreviated name of this bacterium. Is it a synergistic effect between ILTV and Av. paragallinarum? Or a co-infection between them? Please explain why.

L417-418 Please use "eosinophilic intranuclear inclusion bodies".

L438 Please use "fowlpox" instead of "pox"

L465-466 Please use "ILT" instead of "infectious laryngotracheitis" both times.

L469 Please delete "internal organs"

L470 Please use the abbreviated name for this bacteria

L470 Please start the sentence with "In our study,".

L472 Please change from "good" to "strict"

Round 2

Reviewer 2 Report

Dear authors

The authors provided an improved new version of the reviewed manuscript including most of the suggested changes and edits.

The authors must include the provided answer referring to the suggestion of L215 about the permit/code from the local ethics committee for animal welfare in order to handle and euthanize the birds properly. Please consider that without that valuable paragraph a serious ethical concern could be upon this reviewed work. 
